

# Garden centre customer attitudes to pollinators and pollinator-friendly planting

Veronica R. Wignall, Karin Alton and Francis L.W. Ratnieks

Laboratory of Apiculture & Social Insects, School of Life Sciences, University of Sussex, Falmer, Brighton, East Sussex, United Kingdom

## ABSTRACT

Growing nectar- and pollen-rich flowering plant varieties in domestic gardens and other greenspace is an important pro-environmental behaviour that supports pollinating insects. Wildlife gardening is popular in the UK; however, public attitudes and behaviour relating to planting for pollinators are currently not well understood. We investigated these through questionnaires and interviews with customers in five garden centres in Sussex, southeast England, a relevant and useful consumer group representing horticulturally-engaged members of the public. Garden centre customers had strongly positive attitudes and were motivated to plant for bees and other pollinators: most (77%) grew pollinator-friendly varieties, while 64% would be more likely to buy a plant with a pollinator-friendly logo. Personal motivation to support pollinators was linked to a recent increase in personal and public awareness of their declines through (often negativistic) information from mass media sources. Practical implications of these findings in relation to the horticultural retail industry are discussed.

# INTRODUCTION

Growing varieties of flowering plants that support pollinating insects is one of the most effective behaviours through which the general public can directly help these insects, which are considered to be in decline in the UK and worldwide, in part due to reduced availability of nectar- and pollen-producing flowers (*Carvell et al., 2006*; *Potts et al., 2010*; *Vanbergen & the Insect Pollinators Initiative, 2013*; *Sánchez-Bayo & Wyckhuysbc, 2019*). Gardens and other private or community greenspace (e.g., allotments, cemeteries) have been shown to provide an important resource for flower-visiting insects in both rural (*Bates et al., 2011*; *Samnegård, Persson & Smith, 2011*) and particularly urban environments (*Ahrné, Bengtsson & Elmqvist, 2009*; *Gunnarsson & Federsel, 2014*; *Baldock et al., 2019*). Many UK residents engage in wildlife gardening, an increasingly common pro-environmental behaviour (*Gaston et al., 2007*; *Goddard, Dougill & Benton, 2013*). Furthermore, in a survey of over 500 households in Leeds, 41% of participants stated that watching or attracting wildlife was an important reason for using their garden (*Goddard, Dougill & Benton, 2013*). However, public attitudes specifically towards flower-visiting insects and supporting these in gardens

Corresponding author
Veronica R. Wignall,
v.wignall@sussex.ac.uk

or other green space, including through planting attractive flowering plant varieties, has not been assessed to our knowledge.

One indicator that the British public are interested in bees and other pollinators is a high level of recent participation in nationwide pollinator monitoring and citizen science programmes, facilitated over the last decade through technology including widely available smartphone applications. For example, in 2018, 482,915 records of bees were submitted by 23,755 participants in the 'Great British Bee Count' led by Friends of the Earth (UK); 73% of these sightings were made in gardens (*Friends of the Earth (UK), 2018*). Meanwhile, also in the last decade, several online resources to engage and inform the public about gardening for bees and other pollinating insects have been published by sources including Friends of the Earth (UK) (http://www.friendsoftheearth.uk/bees/gardening-bees), the Wildlife Trusts (http://www.wildlifetrusts.org/actions/plant-flowers-bees-and-pollinators) and the Royal Horticultural Society (http://www.rhs.org.uk/advice/profile?PID=648). Being well-informed is a predictor of pro-environmental action (*Easman, Abernethy & Godley, 2018*); therefore, it is possible that this recent increase in availability of online information may have also led to a corresponding growth in public interest in and motivation to plant for pollinators.

Members of the UK public commonly purchase plants from garden centres, horticultural retail outlets that sell plants and gardening material. British garden centre customers spent £1.4 billion on garden plants in 2016 (*Horticultural Trades Association, 2017*) and two thirds of adults visit a garden centre at least once a year (*Horticultural Trades Association, 2018*). Customers in garden centres represent a sample of the UK public that have an interest in gardening, many of whom are likely regularly to plant ornamental flowering plants to varying extents, or have the potential to do so. Therefore, this customer group is relevant and useful to understand the attitudes and behaviours of horticulturally-engaged members of the public relating to pollinators and pollinator-friendly planting. Improving our understanding of this through quantitative and qualitative investigation is an important step in improving floral resources for pollinators.

Since garden centres are a major source of ornamental flowering plants to the general public, it is also likely that increasing the availability and signposting of pollinator-friendly varieties could have a direct positive impact on resource availability for pollinators throughout the UK. However, one recent study revealed that many flowering plants on sale in garden centres were not attractive to flower-visiting insects, in some instances even when labelled as pollinator-friendly (*Garbuzov, Alton & Ratnieks, 2017*). A second recent study identified pesticides in the nectar and pollen of a large proportion of 'bee-friendly'-labelled plants sampled in garden centres, in some cases at levels known to cause harm to bees (*Lentola et al., 2017*). This suggests that garden centres are not currently fulfilling a significant potential to facilitate pollinator-friendly planting. The garden retail industry is influenced by socio-cultural drivers including consumer pro-environmental attitudes and behaviour (*Horticultural Trades Association, 2017*), therefore clarifying customer attitudes towards pollinators could have an important practical implication in respect to the garden centre industry.

This study investigates the attitudes of customers in garden centres towards pollinators and towards growing and purchasing plants that support flower-visiting insects. Our methods simultaneously assess whether there is scope for garden centres to play a more active role in facilitating pollinator-friendly planting. We collected questionnaire responses from 150 visitors to five garden centres in Sussex, southeast England. The questionnaire gathered information about (i) public attitudes to wildlife including pollinators and (ii) existing pro-environmental behaviours relating to pollinators and knowledge about pollinator-friendly plants, including awareness of plant labelling and information provided by garden centres. This was followed up with 14 in-depth interviews with additional customers, to explore selected findings in more detail using a qualitative research approach. Possible implications of the study findings are discussed, including practical application in the garden retail industry.

## MATERIALS AND METHODS

### Garden centres

With permission from the managers, we gathered information from customers visiting five garden centres in Sussex, England using questionnaires and interviews. These were typical of the area, of similar sizes, and included both independent businesses ($n = 2$) and branches of larger chains ($n = 3$). All seemed to have a similar customer base with no notable differences in exclusivity or 'high-end' nature.

### Questionnaire design

The questionnaires had three sections gathering (i) complementary information on the customer (age, sex, reason for visit etc.), (ii) attitudes to wildlife including pollinators and existing pro-environmental behaviours, and (iii) awareness of and attitude towards pollinator-friendly plants, including plant labelling and information provided by garden centres. There was space at the end for comments (Appendix B).

### Garden centre visitor questionnaires and interviews
*Questionnaires*

In total, 150 questionnaires were completed, 30 per garden centre, in August and September 2018, with data gathered on one or two days mainly on weekdays (Table 1).

Customers were approached in the areas with plants for sale and asked if they would be happy to take part in a research study for the University of Sussex. In order not to influence the responses, researchers did not mention pollinators, wildlife, plants or anything relevant to the study, nor did they answer any questions about these topics while the participant was filling out the questionnaire. If asked any questions, we explained that we had an information sheet on the project to give them once they had completed the questionnaire.

We found no significant difference in either the proportions of male and female customers (Chi-squared test, $\chi^2_{(1)} = 0.106$, $P = 0.745$) or the representation of different age groups (Fisher's exact test, $P = 0.905$) between independent and chain stores, so all 150 questionnaire responses were pooled for analysis. For certain questions respondents who
**Table 1** Details of five garden centres where questionnaires and interviews were conducted.

| Garden Centre | Surveys | Interviews |
| --- | --- | --- |
| Wyevale[a] *Newhaven Road, Kingston, Lewes, BN7 3NE* | *n* = 30; 5 September 2018 | – |
| Hillier[a] *Hailsham Road, Pevensey, BN24 5BS* | *n* = 30; 15 August 2018 | – |
| Notcutts[a] *Common Ln, Ditchling, Hassocks, BN6 8TN* | *n* = 30; 7 & 8 August 2018 | *n* = 14; 1, 2 & 5 October 2018 |
| Stavertons Nursery[b] *Eastbourne Road, Halland, BN8 6PU* | *n* = 30; 21 August 2018 | – |
| Rushfields Plant Centre[b] *Poynings, Brighton, BN45 7AY* | *n* = 30; 13 & 15 September 2018 | – |

**Notes.**
[a] Three were branches of a larger chain of garden centres.
[b] Two were independents with only a single garden centre.
All were in Sussex. Dates of customer surveys and interviews are shown by location.

ticked an incorrect number of boxes were removed from the dataset, resulting in some question sample sizes of <150. Sample sizes for questionnaire responses were 150 unless noted otherwise in the text (Results).

### Interviews

After we had reviewed the questionnaire responses, we conducted 14 semi-structured interviews with separate customers (who had not previously completed the questionnaire) in October 2018 in one garden centre, to provide further insights where our findings were interesting and/or led to further questions (*Goddard, Dougill & Benton, 2013*; Table 1). We based the interviews in Wyevale garden centre, Lewes, as this is a branch of a popular large chain and was thought to have a comprehensively representative customer demographic (Table 1). We approached customers browsing in the garden centre and asked if they would be happy to spend 10–15 min answering some informal questions for a research project, in exchange for free refreshments.

Interviewees were either in a pair (*n* = 11 pairs, 22 people) or single (*n* = 3 people). The interviewer (VW) informed the participant(s) that they would be recorded, and each was asked to read and sign an information/consent form before the interview began. Each interview had three sections (Appendix D). In Section 1, we asked the participant(s) to complete one customer questionnaire. If they were a pair, we asked the keener gardener of the two to answer the questionnaire. These 14 questionnaire responses were not included in our analysis of the 150 questionnaires completed previously. In Section 2, we asked for further details on their responses to some of the questions (Q 6, 8, 9, 11, 12, 13, 14, 15, 16). In Section 3, we asked each interviewee (*n* = 25) two further questions not related to the questionnaire (Q + 1: *Has your awareness of/interest in bees and other pollinators/pollinator-friendly plants changed over time? If so, could you tell me a little more about this?* and Q + 2: *Where do you think you receive most information about pollinators?*).

Transcripts were manually analysed using qualitative inquiry (*Saldaña, 2013*). Themes were drawn out using both *in vivo* and descriptive coding, to extract the most appropriate content and essence of the interviews (*Saldaña, 2013*). After organising themes into categories and subcategories, these were cross-referenced against quantitative survey findings and integrated within these themes in the Results section.

### Ethical approval and garden centre permissions

In each garden centre we obtained the manager's permission to survey customers on the premises. On arrival we let the staff know that we were surveying customers on that day.

All survey materials were approved by the University of Sussex Sciences & Technology Cross-Schools Research Ethics Committee (C-REC, project reference number ER/VW58/4). Interview transcriptions and corresponding signed consent forms were given unique reference codes and stored separately under password so that customers could withdraw their consent if they wished. The customer questionnaire and information sheet, and interview questions are available in the Appendices B–D.

### Pollinator-friendly logo size on plant labels

In three garden centres, including one in the five used for questionnaires (Rushfields Plant Centre, Sussex) plus two additional (Brighton Wyevale, Sussex; Gates Garden Centre, Rutland), we surveyed the pollinator-friendly logos present in the plant and bulb stock displayed at the time, as well as seed packets on display (October 10, 2018). This was not to make a comprehensive record of the logos used but provide additional information relating to Q15 in the questionnaire *Do you think the [pollinator-friendly] labels are visible enough?* by measuring the size of a representative sample of pollinator-friendly logos found on labels and packets in the three centres as a proportion of the size of the overall label/packet.

We photographed any plant labels and bulb packets that included a pollinator-friendly logo with a ruler for scale. As there were very large numbers of seed packets we haphazardly selected ten packets with a pollinator-friendly logo for measurement.

We found eight different pollinator-friendly logos at the time of our surveys in the three garden centres (Fig. 1). The most commonly observed was the RHS (Royal Horticultural Society) Perfect for Pollinators (Fig. 1H). Since this logo was much more commonly seen than the others, we photographed a representative selection of plant labels that included it, including different growers and label designs ($n = 35$). In order to ensure other logos were represented, we made a deliberate effort to find and photograph these. As such, the sample we collected does not reflect a proportional distribution of logo types on the plant labels displayed in the centres at the time. Sample sizes of the seven other logos found on plant labels were small (Figs. 1A–1G, $n$: $a = 2, b = 2, c = 1, d = 3, e = 2, f = 3, g = 1$).

Logo area and total label size (measured as the visible part of the label, including any text that directly accompanied the label were then measured using ImageJ (version 1.51, 2015). We also noted whether there was any mention of pollinators on the reverse side of the label.

### Statistical analysis

Contingency tests were used to compare the proportions of questionnaire respondents that chose certain flowering plant features and those that were familiar with pollinator-friendly logos (male vs female; interviewees vs overall questionnaire). When all values were >5, we used Chi-squared tests, with a Yates continuity correction if any values were <10 (Yates, 1934).

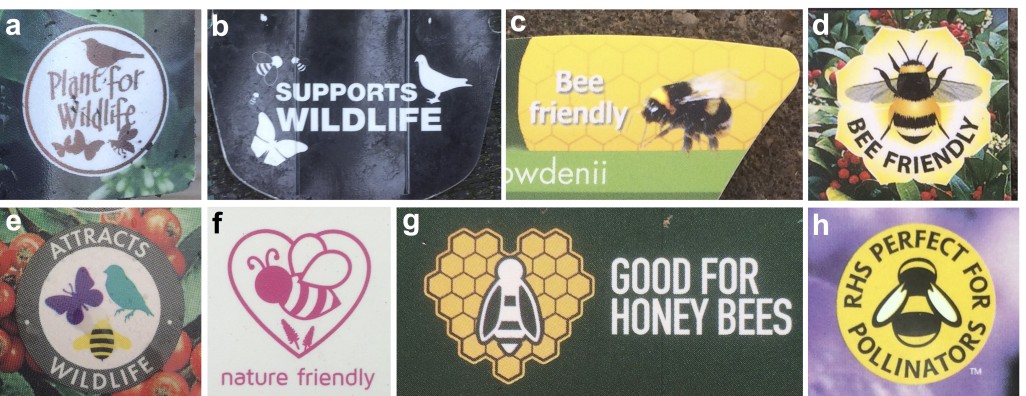

**Figure 1** **Eight wildlife-friendly plant logos (A–H) found on plant labels in three garden centres.** Photo credit: Veronica R. Wignall.

We analysed whether there was any difference in average logo size as a proportion of the total label/packet area between plants, bulbs and seeds using a Kruskal-Wallis rank sum test since data did not fit a parametric distribution.

All statistical analyses were performed using R Studio Version 1.1.463.

## RESULTS

### Questionnaire respondents' characteristics

The majority of the 150 questionnaire respondents were over the age of 55 (78%); most were female (79%; Fig. 2).

Almost all respondents had a garden (95%). When buying plants, 68% most often looked for ornamental plants with flowers ($n = 138$; 12 replies that had incorrectly ticked >1 box for this question were removed), followed by trees or shrubs (20%), vegetable/fruit plants (7%), and lastly indoor plants (5%).

In a multiple response question asking why participants were visiting the garden centre that day, the most common reason was to buy plants or seeds/bulbs (57%), followed by leisure purposes, for example browsing or visiting the cafe (52%). Others were visiting to buy other gardening items such as tools (22%) or non-gardening items (16%).

### Customer attitudes towards wildlife and pollinators (Q 7, 8, 9, 10)

Most questionnaire respondents showed a positive interest in wildlife, with 146 (97%) answering that the decline of wildlife in Britain concerns them. Most did something in their garden or other outside space to help wildlife (97%).

In terms of pollinators specifically, almost all questionnaire respondents (97%) thought that bees and other pollinators were beneficial to their garden or other outdoor area. Most carried out several of five pollinator-friendly actions listed in the questionnaire (mean ± SD = 2.55 ± 1.20 actions, range = 0–5). The most common was to grow pollinator-friendly plants (77% of participants), followed by using limited or no pesticides (64%), providing flowers throughout the year (57%), leaving some areas unmown/unmanaged (37%) and

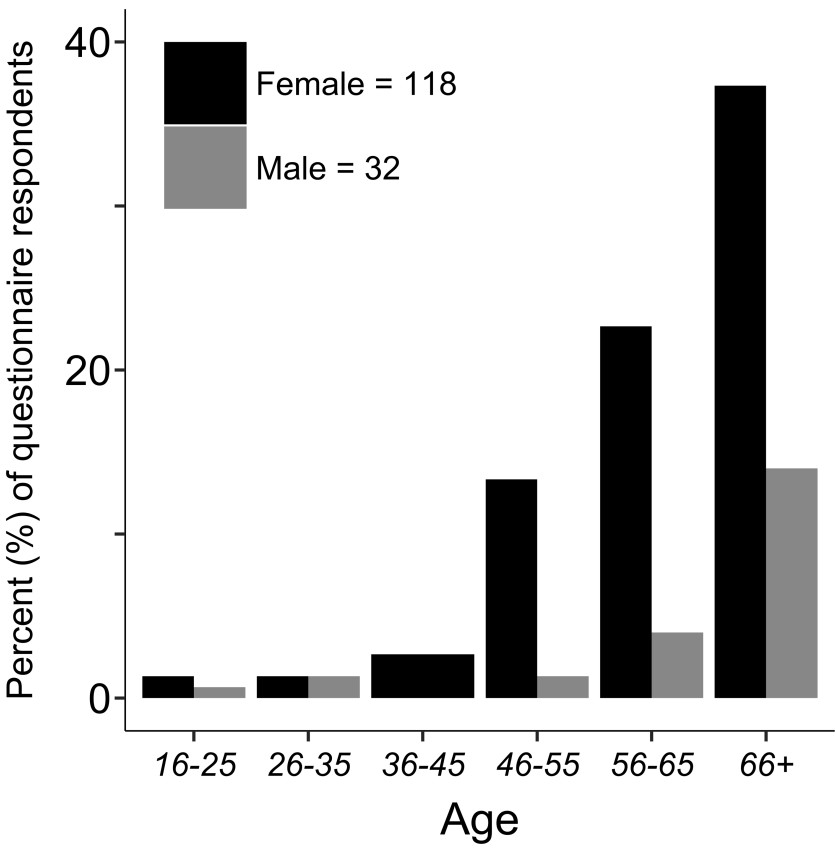

**Figure 2  Questionnaire respondent characteristics.** Age and sex distribution of garden centre customers who answered the questionnaire ($n = 150$).

putting up bee hotels (19%). Only four people said they did not currently help bees and other pollinators in their garden or outside area (Fig. 3A).

Four interviewees mentioned that they disliked wasps. However, in general there was a positive interest in pollinators that was often particularly focused on bees and butterflies. Many interviewees even seemed to feel a psychological benefit of seeing bees and other insects in their garden or outside area, with comments including: "*I was very happy because I got a bees nest in my compost and I liked that*", "*it can be quite therapeutic to sit and watch them [bees]*" and "*I think bees are very important, well I know bees are very important, and we like watching the bees*". As well as this, there was a sense of a positive feeling towards environmental stewardship, with comments such as: "*you just think if it's keeping the natural balance of the ecosystem then it's a good thing*"; "*I love wildlife, I love the bees, I feed the bees, and anything to help nature is better*".

Interviewees also expressed concern for the wellbeing of pollinators, linking this to human and planetary health. One commented "*if we lose our bees, everything else follows suit, so it makes sense to wake up, and you know, start doing more to protect the environment, down from plastic to everything*", another said "*if we run out of bees, if the bees die we die, if*

*they don't pollinate our flowers and our shrubs and our fruits"*, and a third remarked *"put it this way, if the bees go the humans go"*.

### Customer attitudes towards pollinator-friendly planting (Q 6, 14)

Bee- or pollinator-friendly (53%, $n = 145$ replies) was one of three most and equally-important features, excluding price, considered when buying flowering plants, with length of flowering (55%) and hardiness/low level of maintenance (56%). There was no significant difference among these three responses (Chi-squared test, $\chi^2_{(2)} = 0.574$, $P = 0.754$).

Many of the 150 questionnaire respondents said that if a plant has a 'pollinator-friendly' logo on the label they would be more inclined to buy that plant (64%). Almost a third of respondents answered that they would "maybe" be more inclined to buy a plant with a pollinator-friendly label (32%); only six customers (4%) answered that they would not (Fig. 3B).

In the interviews, most of the participants answering the questionnaire (13/14) also stated that they would be more inclined to buy a plant that had a pollinator-friendly label. This might depend on their original purchasing motive, for example: *"I'd only buy it if it fell into my reasons for buying the plant for that space at that time of year. But if it was a choice of two that were equally..., I mean obviously you'd buy the pollinating one"*; and in another interview *"if it was between two [plants] of the same colour and one was pollinating one then I would go for the pollinator-friendly one...I might not actually but I would be tempted to"*. Several interviewees referred to a pollinator-friendly logo as an *"added benefit"* or *"bonus"* that might make them more inclined to purchase a pollinator-friendly plant ($n = 4$ interviews). For example: *"We know what we like, but if it says that on there then it's a bonus"*.

For other interview participants, the presence of a label would either assist their purchasing decision (*"if that label was on one of the...[plants] it would help me choose"*; *"if I was looking at two plants and I couldn't make up my mind, then I would possibly go for the one that had that on [rather than] the other one didn't"*) or provide a clear motive to buy one plant over another, for example: *"When I look through the catalogue I always look to see what all the little symbols are, and if it's a bee-friendly one, definitely if it's a bee-friendly one I think I can justify buying it"*.

### Perceived barriers to pollinator-friendly planting

Interviewees identified certain barriers to planting for pollinators, including allergic reaction to bee stings: *"I was stung by a bee, so I have to carry an epi-pen...we used to have the big area of wildflowers in the middle of the garden, but we don't have that anymore"*. Concern about children being stung was also discussed: *"if you were asking us ten years ago we'd have been going ' no I don't want bees, I've got three-year olds running round the garden'...I wonder whether younger mums would be more concerned"*.

In one interview, price was mentioned as a potential barrier: *"[we would be more inclined to buy a plant with a pollinator-friendly label] as long as the cost didn't go up because of that, because that's what happens...I think because they're marked as pollinator-friendly, they'd put the price up"*.

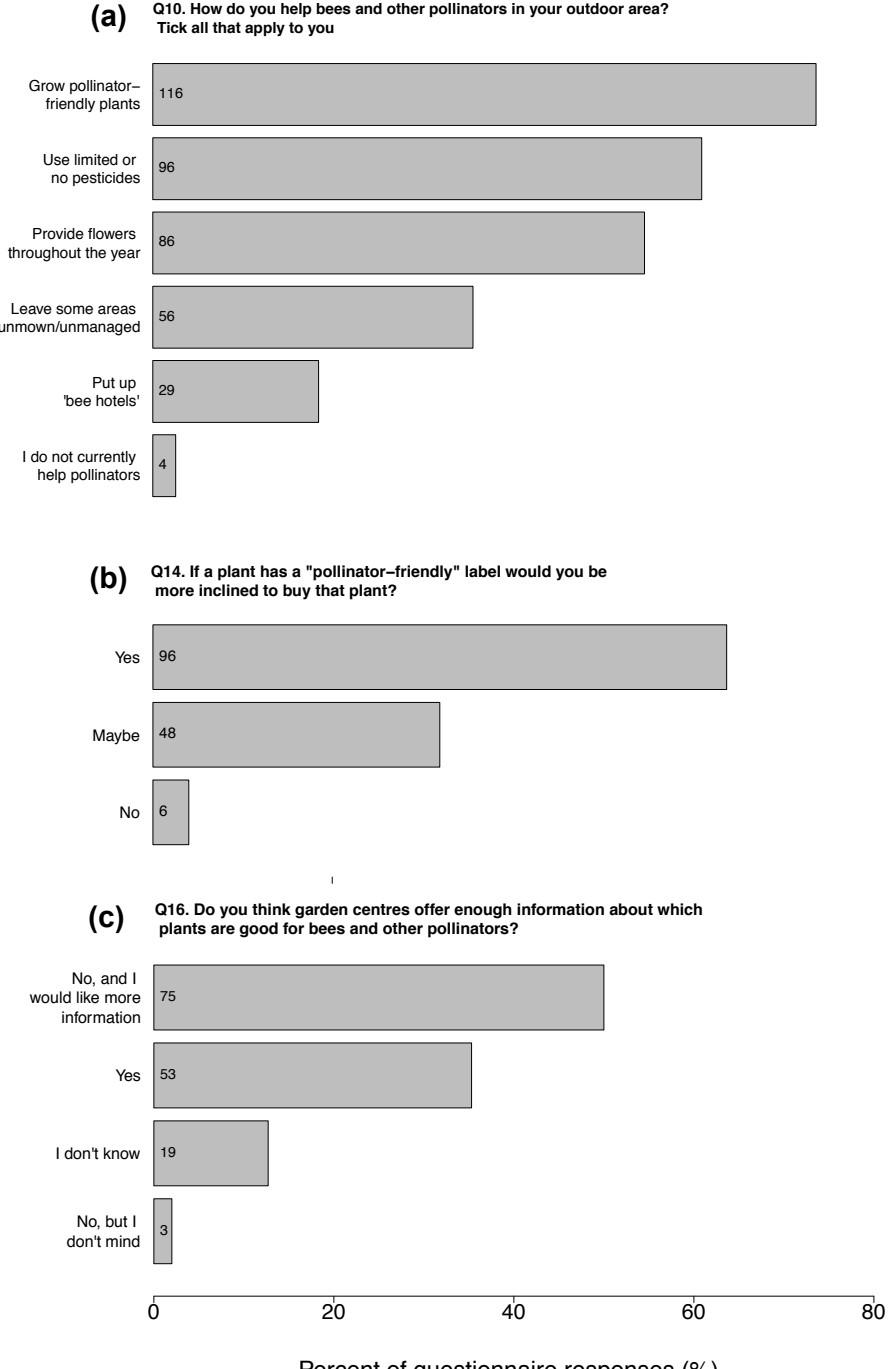

**Figure 3** **Garden centre customers' engagement and interest in pollinator-friendly gardening, compared to their perception of information provision by garden centres.** Grey horizontal bars show the percent of questionnaire responses for each answer to three questions in (A) Question 10 "*How do you help bees and other pollinators in your outdoor area? Tick all that apply to you*", (B) Question 14 "*If a plant has a pollinator-friendly label would you be more inclined to buy that plant*" and (C) Question 16 "*Do you think garden centres offer enough information about which plants are good for bees and other pollinators*". Response details are to the left of each bar. Counts are shown within bars. Sample size was $n = 150$ customers for each question. Questions 10 was a multiple choice question.

## Customer familiarity with pollinator-friendly plant logos (Q 12)

Just over half the questionnaire respondents were familiar with pollinator-friendly plant logos (59%). Proportionally more female participants (F: 64%, n = 75/118) were familiar with the logos than male participants (M: 44%, n = 14/32; Chi-squared test, $\chi^2_{(1)} = 4.094$, $P = 0.043$).

Familiarity with pollinator-friendly plant logos among interviewees was slightly but not significantly lower than in the overall questionnaire (43%, n = 6/14, v 59%, 89/150; Chi-squared test with Yates continuity correction, $\chi^2_{(1)} = 0.830$, $P = 0.362$).

## Customer perception of pollinator-friendly plant logos (Q 13, 15)
### Visibility

Among a subset of 89 questionnaire respondents who were familiar with the logos and were, therefore, able to comment reliably on visibility, 74% thought they were visible enough while 26% did not.

Several interviewees who were familiar with the logos commented that they were noticeable "*if you're looking for them*" (n = 5 of a total of 25 interviewees). For example: "*you have to look for them*"; in another interview "*you know if you're looking for something you're going to see it. If you're not looking for it…*" and in a third "*well sometimes if you're looking, and it's obvious they've got a pollinator-friendly label, well you see it don't you. But I don't always look for it*".

### Reliability

70% of questionnaire respondents answered Yes to Question 13: *Do you think these [pollinator-friendly] labels are reliable sources of information*, despite 28 of these customers having answered that they were not familiar with the logos. 3% did not think they are reliable sources of information, while 27% did not know.

## Pollinator-friendly logo size as a proportion of plant labels

Mean logo size overall was 2.02 ± 1.32% of the total label or packet area (mean ± SD; median = 1.43, n = 65). Proportional area of the pollinator-friendly plant logo was not significantly different between plants (mean ± SD = 2.15 ± 0.21%, n = 49), seeds (1.73 ± 0.26%, n = 10) and bulbs (1.45 ± 0.05, n = 6; Kruskal-Wallis rank sum test, $H_{(2)} = 1.21$, $P = 0.546$; Table 2). The smallest proportional logo type was 1.08% of the total label area ("Good for Honey Bees", n = 1, Fig. 1G) and the largest logo type was 7.58% ("Bee friendly", n = 1, Fig. 1C)).

Most plant labels, seed packets and bulb packets with pollinator-friendly logos on the front did not have any information about pollinators on the reverse of the label or packet (plants: 75%; seeds: 78%; bulbs: 100% (data given for labels for which reverse information was available); Table 2).

## Availability of advice and information in garden centres (Q 16)

Question 16 asked questionnaire respondents whether they think garden centres offer enough information about which plants are good for bees and other pollinators. The most common response was *b. No, and it would be useful to have more information* (50%),

**Table 2** **Wildlife-friendly logo sizes on plant, seed and bulb labels/packets.** Average pooled measurements of pollinator- and wildlife-friendly plant logos and the plant labels ($n = 49$) and packets of seeds ($n = 10$) and bulbs ($n = 6$) on which they were found. Values are given as mean $\pm$ SEM. Any mention of pollinators on the label or packet reverse is indicated for those labels/packets for which this information was available. Logo types (A–H) are shown in Fig. 3.

| | n | Logo types present | Mean logo area (mm$^2$) | Mean total label/packet size (mm$^2$) | Mean logo area as proportion of mean total label/packet area (%) | Mentions pollinators on label/packet reverse? |
|---|---|---|---|---|---|---|
| *Plants* | 49 | a, b, c, d, e, f, g, h | 168.0 ± 16.9 | 8,495 ± 663 | 2.15 ± 0.21 | Yes = 12/47 |
| *Seeds* | 10 | a | 216.5 ± 36.6 | 12,291 ± 387 | 1.73 ± 0.26 | Yes = 2/9 |
| *Bulbs* | 6 | a | 435.2 ± 56.0 | 30,377 ± 4,188 | 1.45 ± 0.05 | Yes = 0/6 |

followed by *a. Yes* (35%). Just three answered *c. No, but I don't mind* (2%), and 19 answered *d. I don't know* (13%; Fig. 3C).

Interviewees answering the questionnaire also most commonly answered *b. No, and it would be useful to have more information* (64%, 9/14). Several commented on the potential for garden centres to provide advice and information about pollinator-friendly plants. One interviewee commented: "*It's probably a place where to start, the garden centres, because it's probably where people go and buy their plants, apart from markets and things.*" For another: "*We love a garden centre don't we, so I mean, well it's the best place to have it really isn't it*", and a third noted "*it's when you're buying the plants that you're thinking about pollinators, I mean not when you're sitting in your sitting room*".

A common theme was the lack of available information in garden centres, with several interviewees making comments similar to this example: "*I've never really walked round the garden centre and seen anything about it*". Some mentioned that the information might be there if you looked for it or had a predetermined interest, for example: "*It depends whether you want to come in and you're interested in it or not, and if you're not, you're just going to go round the garden centre buying the things that you want to buy*".

Many felt that larger displays would be useful both to interest customers in pollinator-friendly plants and to provide information. One interviewee remarked "*I notice when the garden centre has a special section for bee friendly, but I can't say I'm looking for logos*" and another that "*I think you'd have to have it with some sort of big bee display for you to actually whilst you're chatting and looking and kids and stuff, you'd have to have a reason to look at that section*". One interviewee noted the value of larger displays to "*make information more prominent for older eyes. Just to make people aware, just to bring the awareness, that's the main thing.*"

Several made suggestions such as leaflets, displays, guides, posters and grouping plants in a 'pollinator-friendly' section to provide information and attract customer interest.

### Current media interest in pollinators

The most common source of interviewees' knowledge and awareness about pollinators was the media, with one or more of television, 'the news', newspapers and radio mentioned in 12/13 interviews in which the participant(s) were aware of pollinators. Several cited nature programmes, documentaries and/or the popular weekly BBC television program '*Gardener's World*' while other sources included gardening books, education while growing

up and magazines. Social media was also acknowledged by one interviewee who received some of her knowledge from Facebook. Only two of a total of 25 interviewees said they had received any information about pollinators and/or pollinator-friendly plants in garden centres, despite many participants being regular visitors.

We asked interviewees whether they felt there had been any general change in awareness about pollinators over time. One had not been aware of pollinators prior to the interview, and in one interview neither person in the pair gave a clear response to this question. One couple did not feel their awareness of pollinators had changed. However, in 11 interviews, the participant(s) felt there had been a recent increase in the quantity and availability of information about pollinators. One said: "*Definitely, in the last four, five years, there's been newspapers, television, documentaries about it*" while another commented "*There's so much on the TV now, particularly on sky channels and wildlife channels. It's everywhere*".

Despite this, interview participants commented on a lack of reliable, comprehensive information: "*you've got to get your shock headlines out there to talk to people, but often there's not enough back up information, or you've really got to make a concerted effort to go find out why and what and how we can do anything about it*". Several interviewees commented on how the news media can be transient and unreliable, with comments such as: "*now the whole buzz thing has gone to plastic, and all of a sudden the bee awareness has just been pushed aside a bit*"; "*Occasionally there's some news, it comes up on the news about bees and the loss of bees, but then it's all a one-day wonder*". It was commonly noted that the public are often exposed to conflicting information through the news, including about bees and other pollinators, "*…so you think, well I don't really know what the real story is*".

## DISCUSSION

Our results show that UK garden centre customers have a strongly positive attitude towards gardening for pollinators. Almost all (97%) questionnaire respondents thought that bees and other pollinators were beneficial to their garden, most (97%) reported that they already take some action to help these insects in their outside area, and many (53%) prioritised pollinator-friendly features when purchasing flowering plants. These overall conclusions were reinforced through in-depth interviews. This is the first time to our knowledge that positive attitudes towards wildlife gardening in the UK has specifically been shown to include pollinating insects; although many British households actively encourage wildlife in their garden, this often chiefly involves feeding birds (*Gaston et al., 2007*; *Department for Environment Food & Rural Affairs, 2009*).

Positive attitudes towards pollinators and pollinator-friendly plants is likely to influence consumer behaviour (e.g., *Wollaeger, Getter & Behe, 2015*; *Rihn & Khachatryan, 2016*). This may explain why, in our study, 96% of questionnaire respondents answered that they would ('yes' 64%; 'maybe' 32%) be more inclined to buy a plant if it had a 'pollinator-friendly' label (Fig. 3B). Insights from the interviews showed that, for some customers, knowing a plant was good for pollinators would justify their purchase or motivate them to buy a particular plant. For others, a purchasing decision would depend on initial reasons

for buying a plant, but knowing one was pollinator-friendly would help them to choose between, for example, two similar varieties.

In terms of practical actions to support pollinators, 77% of questionnaire respondents stated that they currently grow pollinator-friendly plants (Fig. 3A), although it was not clear whether they had initially acquired these plants with the intention of supporting pollinators; it is possible that this was often a by-product of varieties initially planted for other reasons. However, 53% of respondents considered bee- or pollinator-friendliness to be one of the three most important features, excluding price, when buying flowering plants. A clear incentive to help bees was also shown by a fifth of respondents who put up bee 'hotels'. These structures aim to provide nesting habitats for solitary bees, although their efficacy is unclear (*MacIvor & Packer, 2015*). Many people also gardened with limited or no pesticides. While this is a common pro-environmental behaviour that may reflect consumer awareness of pesticides' negative effects on pollinating insects (*Campbell, Khachatryan & Rihn, 2017*), participants' motivation for this action was not investigated further in this study.

Public action to conserve pollinators is considered a necessary response to pollinator declines (*Cambridge Institute for Sustainability Leadership et al., 2017*). Encouraging citizen action and education forms a major part of the EU Pollinators Initiative (*European Commission, 2018*) and several national-level pollinator strategies (*Senapathi et al., 2017*). Growing pollinator-friendly varieties of flowering plants in gardens and other private or community greenspace is one of the most effective ways in which the general public can directly help flower-visiting insects. Flower availability in both urban and countryside areas can often be reduced due to factors such as a high proportion of impervious surfaces (*McKinney, 2002*) or intensive farming (*Brassley, 2000*; *Ollerton et al., 2014*), whereas gardens can be relatively flower-rich, contain a high diversity of plant species, and even provide a resource at times of the year when native flowers are not in bloom (*Smith, Warren & Thompson, 2006*; *Stelzer et al., 2010*; *Baldock et al., 2015*). Optimising the supply of nectar and pollen in domestic gardens and other greenspace through choosing plant varieties that attract insects (*Garbuzov & Ratnieks, 2014*) is therefore increasingly important in alleviating pollinator dietary stress, particularly since these areas comprise a relatively large total area in the UK (*Gaston et al., 2005*).

Garden centres are well-placed to facilitate this both through supplying plant varieties that will attract flower-visiting insects, and by delivering relevant advice and information to a substantial customer base. Two thirds of British adults visit a garden centre at least once a year (*Horticultural Trades Association, 2018*), and garden centre customers in Great Britain spent £1.4 billion on garden plants in 2016 (*Horticultural Trades Association, 2017*). Here, most respondents were visiting the garden centre to purchase plants, seeds or bulbs (56.7%), and when buying plants most respondents looked for ornamental plants with flowers (68.1%).

Despite this, evidence from this and previous research suggests that the potential for garden centres to facilitate pollinator-friendly planting is not being met, despite clear customer interest (Fig. 3). For example, it is possible for garden centres to use peer-reviewed scientific evidence to select and market varieties of flowering plants that attract pollinators
(*Garbuzov & Ratnieks, 2014*). However, many flowering plants on sale in garden centres are in fact not attractive to flower-visiting insects, in some instances even when labelled 'pollinator-friendly' (*Garbuzov, Alton & Ratnieks, 2017*).

In this study, most customers perceived garden centres' provision of advice and information about pollinator-friendly planting to be limited (Fig. 3C). The majority of questionnaire respondents thought that garden centres did not offer enough (52%) and only two interviewees had received any of their knowledge or information on this topic from garden centres compared to other sources such as television, news media and nature programmes, which were cited several times. This contrasted with a clear desire for more information, since half of all questionnaire respondents thought that it would be useful for garden centres to offer more information. Several interviewees even commented that garden centres would be the "best place" for advice about which plants are attractive to pollinators since this is most useful in context, such as when people are buying plants. Just over a third of questionnaire respondents thought there was enough information in garden centres; of these, 72% were familiar with 'pollinator-friendly' logos. This could indicate that a proportion of customers are generally well-informed on this issue, or alternatively that customers who answered that garden centres do offer enough information are basing this on the occurrence of pollinator-friendly logos.

Pollinator-friendly logos are one way in which garden centres advise customers about which plants are good for pollinators. These 'eco-labels' can be successful marketing tools. Eye-tracking technology has shown that customers who spent time looking at a pollinator-friendly label on a plant were more likely to purchase it than those who did not view the label (*Khachatryan et al., 2017*). In our study the majority of questionnaire respondents stated that they would be more likely to buy a plant that had a pollinator-friendly logo (64%). However, 41% were not familiar with such logos. The logos tend to be small: here, mean pollinator-friendly logo size on plant, bulb and seed labels was just 2.02% of the overall label or packet size, which may explain why many respondents were not familiar with them.

Most respondents who were familiar with pollinator-friendly logos thought that they were visible enough, possibly simply due to the fact that they had seen them. A number of interviewees commented that these logos are noticeable if you are looking for them. This is consistent with previous work investigating the potential efficacy of incentives for residential wildlife gardening, in which interviewees commented that you have to "want to know" in order to find relevant information (*Goddard, Dougill & Benton, 2013*). Many garden centre customers, perhaps particularly younger age groups with competing time demands, are likely to have a passive approach to receiving information about which plants are attractive for pollinators, even if they have a positive attitude towards pollinator conservation. This was summarised by one interviewee: "*I think you'd have to have it with some sort of big bee display for you to actually, whilst you're chatting and looking and kids and stuff, you'd have to have a reason to look at that section*".

Since a lack of information has been shown to be a barrier to wildlife gardening here and in previous research (*Goddard, Dougill & Benton, 2013*; *Campbell, Khachatryan & Rihn, 2017*), this highlights a need for highly visible, accessible information to supplement

pollinator-friendly logos on plant labels. Interviewees suggested a range of options to provide information and attract customer interest, including leaflets, displays, guides and posters. Several mentioned that grouping plants in a 'pollinator-friendly' section with corresponding information would be helpful. Further suggestions based on our findings could be to (i) increase the size of pollinator-friendly logos to make them more visible and (ii) include practical information about pollinators in combination with these logos, since only a small proportion of plant, seed and bulb labels and packets with logos had any mention of pollinators on the reverse of the packet (22.6%; Table 2).

Customers spend a substantial length of time in garden centres, and under 10% of the UK spend on garden plants is made online (*Horticultural Trades Association, 2017*). Unlike many other industries where online retail success has caused traditional stores to be non-viable, the experience of visiting a garden centre to purchase plants is clearly important to customers. This opens the possibility for garden centres to provide obvious, accurate information about pollinators that is available in context and at point-of-sale when customers are buying flowering plants. In this study interviewees noted the usefulness of displays about other aspects of plant qualities and care. It is possible that displays about pollinator-friendly plants could be easily integrated into such pre-existing information infrastructure without significant cost, which might provide a barrier to customers should it be reflected in pollinator-friendly plant prices (*Campbell, Khachatryan & Rihn, 2017*; this study).

We found that interviewees often spontaneously mentioned a positive emotional state associated with seeing bees and other pollinators in their gardens. Gardens and other private outside areas, including allotments, balconies and patios, can provide an important connection to nature, particularly for people living in urban environments (*Dunnett & Qasim, 2000*; *Freeman et al., 2012*; *Cox et al., 2017*). The benefit of wildlife gardening to personal psychological wellbeing has previously been reported (*Goddard, Dougill & Benton, 2013*), and a link between pollinators and emotional wellbeing in this study suggests that this may partly explain a personal motive for gardening for pollinators.

Interestingly, many interviewees reported a recent increase in personal and public awareness of pollinators, which was largely linked to a growth in the quantity of information published in print and broadcast media. This was often negativistic, with several participants mentioning 'shock' or 'dramatic' headlines, the need to 'look after' pollinators such as bees and genuine concern about their declines (recently reviewed in *Sánchez-Bayo & Wyckhuysbc, 2019*). This is similar to a recent survey of environmental professionals and members of the British public which found a large proportion of participants gained their information from mainstream media sources (*Easman, Abernethy & Godley, 2018*). In this study, individuals that were more concerned about the marine environment were more likely to engage with pro-environmental actions to minimise their personal impact (*Easman, Abernethy & Godley, 2018*). It is likely that higher awareness of and concern for pollinators would contribute to gardeners' personal motivation to encourage pollinating insects. Concern for the status of pollinators may also have added to reported feelings of happiness associated with seeing them in their outside area, since humans disproportionately value

rarity, which has been linked to increased interest in rare and threatened animal species (*Angulo & Courchamp, 2009*).

Awareness of pollinators and factors associated with their declines can influence plant purchasing decisions based on pro-environmental attributes. In one study, consumers who were aware of neonicotinoid pesticides, which have gained widespread media attention due to their negative effects on insect pollinator health (reviewed in *Van der Sluijs et al., 2013*, were significantly more likely to buy plants labelled 'neonic-free' than those who were not aware (*Rihn & Khachatryan, 2016*). A taste for sustainable products has been identified as a major socio-cultural driver in the garden centre retail industry by the Horticultural Trade Association (HTA). For example, it is becoming important to meet a growing demand for alternatives to plastic and peat, materials considered to be environmentally unsustainable, due to increasing customer antipathy (*Horticultural Trades Association, 2017*). Here, several interviewees described pollinator-friendly qualities as an 'added bonus' to plants they might purchase primarily for other reasons. While this is a positive step, it also suggests that more could be done to harness a clear motivation of garden centre customers to support pollinators, for example by specifically advertising pollinator-friendly features to drive sales of these plants. Investigating whether featuring pollinator-friendly qualities as a primary attraction would increase sales of these plants compared to (i) the same but un-labelled pollinator-friendly or (ii) similar non-pollinator-friendly varieties is a logical next step, since this could show the empirical value of this type of marketing for garden centres themselves.

## CONCLUSIONS

Growing pollinator-friendly varieties of flowering plants is one of the most effective ways in which members of the public can directly help bees and other pollinators, which are known to be in decline in the UK (e.g., *Carvell et al., 2006*) and globally (*Potts et al., 2010*; *Sánchez-Bayo & Wyckhuysbc, 2019*). It is therefore important to understand public attitudes towards planting for pollinators; however, this has not been directly studied as far as we are aware. This study investigates garden centres customers' attitudes towards pollinators and pollinator-friendly planting, since this relevant and important consumer group represents members of the UK public who are actively engaged in gardening. We show for the first time that customers have, in general, a strong current interest in and positive attitude towards pollinating insects, which translates into an impetus to plant pollinator-friendly plant varieties in private gardens or other outdoor areas. Facilitating this could have a real impact on provision of floral resources for pollinating insects, since gardens make up a large area of the UK (*Gaston et al., 2005*), and are increasingly important sources of nectar and pollen for pollinators particularly in urban areas (*Baldock et al., 2019*).

We also suggest that our findings are relevant to the horticultural retail industry, since provision of evidence-based advice and information about pollinators and pollinator-friendly planting, as well as promotion of such plants, could potentially be increased without substantial involved costs to garden centres. We speculate that this would be likely to benefit sales due to a strong customer interest, although this deserves further study; as well as having a positive effect on the pollinators themselves.

## ACKNOWLEDGEMENTS

The authors would like to thank the management team of each garden centre in this study, particularly Paul Cooke (Wyevale, Lewes), Tom and Phil Cottingham (Staverton Nursery), Michael Edmondson (Hillier, Hailsham), Kathryn Hillman (Rushfields Plant Centre) and Gary West (Notcutts, Ditchling) for their assistance and kind permission to carry out questionnaire surveys and interviews with customers on their premises. We also thank Thomas Green for his valuable help in data collection.

### Funding

This work was supported by the C.B. Dennis British Beekeepers' Research Trust and the University of Sussex which equally fund Veronica R. Wignall's PhD studentship, of which this study is a part. There was no additional external funding received for this study. The funders had no role in study design, data collection and analysis, decision to publish, or preparation of the manuscript.

### Grant Disclosures

The following grant information was disclosed by the authors:
C.B. Dennis British Beekeepers' Research Trust and the University of Sussex.

### Competing Interests

The authors declare there are no competing interests.

### Author Contributions

- Veronica R. Wignall conceived and designed the experiments, performed the experiments, analyzed the data, prepared figures and/or tables, authored or reviewed drafts of the paper and approved the final draft.
- Karin Alton conceived and designed the experiments, performed the experiments and approved the final draft.
- Francis L.W. Ratnieks conceived and designed the experiments, gave guidance and feedback on original versions of the manuscript, and approved the final draft.

### Human Ethics

The following information was supplied relating to ethical approvals (i.e., approving body and any reference numbers):

The University of Sussex Sciences & Technology Cross-Schools Research Ethics Committee (C-REC) granted Ethical approval to carry out this work (project reference number ER/VW58/4).

### Data Availability

The raw data are available in Dataset S1. Raw data show customer responses (positive answers indicated by a '1') to questions 1–18 of the questionnaire.

## Supplemental Information

Supplemental information for this article can be found online at http://dx.doi.org/10.7717/peerj.7088#supplemental-information.

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
