# Peer review of "Garden centre customer attitudes to pollinators and pollinator-friendly planting"

_PeerJ, doi:10.7717/peerj.7088_

## Round 0.1 · original submission · Minor Revisions

Thank you for your submission. I look forward to receiving a revised manuscript. I believe by incorporating the comments from reviewer 1, some of the logo analyses issues reviewer 2 mentions should be addressed. Please consider all comments and clearly address if they were or were not incorporated.

Reviewer 1 ·

Basic reporting

No comment - all fine.

Experimental design

No comment - all fine.

Validity of the findings

No comment - all fine.

Additional comments

• Quite an interesting and timely paper as these topics are often discussed – how many people are trying to plant for pollinators, and how to get people to buy more pollinator-friendly plants. Findings are unsurprising given the audience targeted and previous general knowledge, but it is good to have this documented.
• Line 113 – you say the garden centres are representative of the area; are they representative in terms of socio-economic status of the customers and availability and cost of plant stock? I.e. are any of them more expensive or believed to be more high-end? (you do say later that you had a similar proportion of males: females and ages but this is another angle)
• Line 199 – how many logos in colour vs black and white; stylized vs real photos; have text besides logo and title?
• Line 238 – what was the sample size for this question? And following ones? Is it always 150 unless otherwise specified?
• Line 277 – do you have any data on what is actually being purchased vs what they say they prefer to purchase? i.e. do respondents say they purchase for pollinators when asked via the survey ‘select all’ responses, but do they actually do so in reality? Ditto for Line 282 – testing similar varieties/appearances of plants with and without the labels? (you get into it briefly in the interviews/Line 287 area, but any more information than that)?
• Line 307 – do you need to perhaps clarify better in the paper questionnaire respondents vs interviewees, when you discuss the results? Technically respondents are also interviewees. When I hit this example I paused with a bit of minor confusion then was find for the other uses.
• Line 460-462,488 – why didn’t you discuss the bee hotels & pesticide usage, and the reasons for visiting in the results section? (or did you?)
• Line 498 paragraph – repeating results here again; can you summarize rather than repeating yourself?
• Table 1 caption: you state three garden centres were: a) chains and b) independents. Clarify how many of each type of garden centre was used – it implies here that there were 3 of each, but that is not accurate as previously you said 5. Or perhaps this is a typo, and you mean to say three garden centres, and the letters a and be refer to the table superscripts.
• Table 2: for the last column, does the label/package mention pollinators, why is the total sample size for this calculation not the same as the overall sample size? i.e. /47 vs /49 for plants?
• Figure 1: label the x axis as ages?
• Figure 3: minor comment, consider listing the types (plants, seeds, bulbs) in the same order as you do in Table 2

·

Basic reporting

no comment

Experimental design

I find this to be a relevant and informative piece, but I am uncertain and confused by some of the experimental design and analysis relating to label size and proportion, in particular that in my reading of this piece the size and proportion of the logos are not assessed in any manner that seems to add value to this point. Were larger logo more commonly recognized? Were logos on one type of plant more commonly recognized? This particular analysis doesn't go far enough and there is quite a bit of potential to expand on it. My suggestion is removing this analysis in its current form, or providing some expanded analysis (either with survey data if possible, or an analysis by the authors of the validity of each label). Unless I have missed something in my reading there wasn't an analysis of the validity of the logos by the research group, and this would be useful. Perhaps this is pending in a subsequent publication, but as I have stated before, the current treatment of logo analysis is more confusing than useful. I would be very happy to see the authors provide their own review of the logos, and highlight if there are the perceived information gaps that some of the interviewees alluded to.

Validity of the findings

Following up on my comments above, I think there is room to expand on expert inference and opinion in the discussion, especially with respect to the quality of information available to the public.

As a further follow-up, my suggestion of moving away from the statistics of logo size would leave room to quantify some of the findings of the survey, and present them graphically. I would suggest some additional visual of the coded survey responses to help outline the points made about a strong interest in pollinator gardening combined with the deficit in information.

Additional comments

This is a great piece on public impressions and very relevant to planning conservation actions. I have some concerns about the presentation of findings and am suggesting some more detail be added to further quantify the findings.

---

## Round 0.2 · accepted · Accept

Thank you for clearly addressing the reviewer comments. Congratulations!